# Effective Structural Encodings via Local Curvature Profiles

**Lukas Fesser**
Faculty of Arts and Sciences
Harvard University
lukas_fesser@fas.harvard.edu

**Melanie Weber**
Department of Applied Mathematics
Harvard University
mweber@g.harvard.edu

## Abstract

Structural and Positional Encodings can significantly improve the performance of Graph Neural Networks in downstream tasks. Recent literature has begun to systematically investigate differences in the structural properties that these approaches encode, as well as performance trade-offs between them. However, the question of which structural properties yield the most effective encoding remains open. In this paper, we investigate this question from a geometric perspective. We propose a novel structural encoding based on discrete Ricci curvature (*Local Curvature Profiles*, short *LCP*) and show that it significantly outperforms existing encoding approaches. We further show that combining local structural encodings, such as LCP, with global positional encodings improves downstream performance, suggesting that they capture complementary geometric information. Finally, we compare different encoding types with (curvature-based) rewiring techniques. Rewiring has recently received a surge of interest due to its ability to improve the performance of GNNs by mitigating over-smoothing and over-squashing effects. Our results suggest that utilizing curvature information for structural encodings delivers significantly larger performance increases than rewiring. [1]

## 1 Introduction

Graph Machine Learning has emerged as a powerful tool in the social and natural sciences, as well as in engineering (Zitnik et al., 2018; Gligorijević et al., 2021; Shlomi et al., 2020; Wu et al., 2022). Graph Neural Networks (GNN), which implement the message-passing paradigm, have emerged as the dominating architecture. However, recent literature has revealed several shortcoming in their representational power, stemming from their inability to distinguish certain classes of non-isomorphic graphs (Xu et al., 2018; Morris et al., 2019) and difficulties in accurately encoding long-range dependencies in the underlying graph (Alon & Yahav, 2021; Li et al., 2018). A potential remedy comes in the form of *Structural* (short: *SE*) and *Positional Encodings* (short: *PE*), which endow nodes or edges with additional information. This can be in the form of local information, such as the structure of the subgraph to which a node belongs or its position within it. Other encoding types capture global information, such as the node's position in the graph or the types of substructures contained in the graph. Strong empirical evidence across different domains and architectures suggests that encoding such additional information improves the GNN's performance (Dwivedi et al., 2023; Bouritsas et al., 2022; Park et al., 2022).

Many examples of effective encodings are derived from classical tools in Graph Theory and Combinatorics. This includes spectral encodings (Dwivedi et al., 2023; Kreuzer et al., 2021), which encode structural insights derived from the analysis of the Graph Laplacian and its spectrum. Another popular encoding are (local) substructure counts (Bouritsas et al., 2022), which are inspired by the classical idea of *network motifs* (Holland & Leinhardt, 1976). Both of those encodings can result in expensive subroutines, which can limit scalability in practise. In this work, we turn to a different classical tool: Discrete Ricci curvature. Notions of discrete Ricci curvature have been previously considered in the Graph Machine Learning literature, including for unsupervised node clustering (Ni et al., 2019; Sia et al., 2019; Tian et al., 2023), graph rewiring (Topping et al., 2022; Nguyen et al.,

---

[1] Code available at https://github.com/Weber-GeoML/Local_Curvature_Profile

2023) and for utilizing curvature information in Message-Passing GNNs (Ye et al., 2020; Lai et al., 2023). Here, we propose a novel local structural encoding based on discrete Ricci curvature termed *Local Curvature Profiles* (short: *LCP*). We analyze the effectiveness of LCP through a range of experiments, which reveal LCP's superior performance in node- and graph-level tasks. We further provide a theoretical analysis of LCP's computational efficiency and impact on expressivity.

Despite a plethora of proposed encodings, the question of which structural properties yield the most effective encoding remains open. In this paper, we investigate this question from a geometric perspective. Specifically, we hypothesize that different encodings capture complementary information on the local and global properties of the graph. This would imply that combining different encoding types could lead to improvements in downstream performance. We will investigate this hypothesis experimentally, focusing on combinations of local structural and global positional encodings, which are expected to capture complementary structural properties (Tab. 2.1).

Graph rewiring can be viewed as a type of relational encoding, in that it increases a node's awareness of information encoded in long-range connections during message-passing. Several graph rewiring techniques utilize curvature to identify edges to remove and artificial edges to add (Topping et al., 2022; Nguyen et al., 2023). In this context, the question of the most effective use for curvature information (as rewiring or as structural encoding) arises. We investigate this question through systematic experiments.

## 1.1 SUMMARY OF CONTRIBUTIONS

The main contributions of this paper are as follows:

1. We introduce *Local Curvature Profiles* (short *LCP*) as a novel structural encoding (sec. 3.1). Our approach encodes for each node a characterization of the geometry of its neighborhood. We show that encoding such information provably improves the expressivity of the GNN (sec. 3.2) and enhances its performance in downstream tasks (sec. 4.2.1).

2. We further show that combining local structural encodings, such as LCP, with global positional encodings improves performance in node and graph classification tasks (sec. 4.2.2). Our results suggest that local structural and global positional encoding capture complementary information.

3. We further compare LCP with curvature-based rewiring, a previous approach for encoding curvature characterizations into Graph Machine Learning frameworks. Our results suggest that encoding curvature via LCP leads to superior performance in downstream tasks (sec. 4.2.3).

We perform a range of ablation studies to investigate various design choices in our framework, including the choice of the curvature notion (sec. 4.3).

## 1.2 RELATED WORK

A plethora of structural and positional encodings have been proposed in the GNN literature; notable examples include encodings of spectral information (Dwivedi et al., 2022a; 2023); node distances based on shortest paths, random walks and diffusion processes (Li et al., 2020; Mialon et al., 2021); local substructure counts (Bouritsas et al., 2022; Zhao et al., 2022) and node degree distributions (Cai & Wang, 2018). A recent taxonomy and benchmark on encodings in Graph Transformers can be found in (Rampasek et al., 2022). Notions of discrete Ricci curvature have been utilized previously in Graph Machine Learning, including for Graph Rewiring (Topping et al., 2022; Nguyen et al., 2023; Fesser & Weber, 2023) or directly encoded into message-passing mechanisms (Ye et al., 2020) or attention weights (Lai et al., 2023). Beyond GNNs, discrete curvature has been applied in community detection (Ni et al., 2019; Sia et al., 2019; Fesser et al., 2023; Tian et al., 2023), representation learning (Lubold et al., 2023; Weber, 2020) and graph subsampling (Weber et al., 2017), among others.

## 2 BACKGROUND AND NOTATION

Following standard convention, we denote GNN input graphs as $G = (X, E)$ with node attributes $X \in \mathbb{R}^{|V| \times m}$ and edges $E \subseteq V \times V$, where $V$ is the set of vertices of $G$.

## 2.1 GRAPH NEURAL NETWORKS

**Message-Passing Graph Neural Networks.** The blueprint of many state of the art Graph Machine Learning architectures are Message-Passing Graph Neural Networks (MPGNNs) (Gori et al., 2005; Hamilton et al., 2017). They learn node embeddings via an iterative scheme, where each node's representation is iteratively updated based on its neighbors' representations. Formally, let $\mathbf{x}_v^l$ denote the representation of node $v$ at layer $l$, then the representation after one iteration of message-passing, is given by

$$\mathbf{x}_v^{l+1} = \phi_l \left( \bigoplus_{p \in \mathcal{N}_v \cup \{v\}} \psi_l \left( \mathbf{x}_p^l \right) \right),$$

The number of such iterations is often called the *depth* of the GNN. The initial representations $\mathbf{x}_v^0$ are usually given by the node attributes in the input graph. The specific form of the functions $\phi_k, \psi_k$ varies across architectures. In this work, we focus on three of the most popular instances of MPGNNs: *Graph Convolutional Networks* (short: *GCN*) (Kipf & Welling, 2017), *Graph Isomorphism Networks* (short: *GIN*) (Xu et al., 2018) and *Graph Attention Networks* (short: *GAT*) (Veličković et al., 2018).

**Representational Power.** One way of understanding the utility of a GNN is by analyzing its representational power or *expressivity*. A classical tool is isomorphism testing, i.e., asking whether two non-isomorphic graphs can be distinguished by the GNN. A useful heuristic for this analysis is the Weisfeiler-Lehman (WL) test (Weisfeiler & Leman, 1968), which iteratively aggregates labels from each node's neighbors into multi-sets. The multi-sets stabilize after a few iterations; the WL test then certifies two graphs as non-isomorphic only if the final multi-sets of their nodes are not identical. Unfortunately, the test is only a heuristic and may fail for certain classes of graphs; notable examples include regular graphs. A generalization of the test ($k$-*WL test* (Cai et al., 1989)) assigns multi-sets of labels to $k$-tuples of nodes. It can be shown that the $k$-WL test is strictly more powerful than the $(k-1)$-WL test in that it can distinguish a wider range of non-isomorphic graphs. Recent literature has demonstrated that the representational power of MPGNNs is limited to the expressivity of the 1-WL test, which stems from an inability to trace back the origin of specific messages (Xu et al., 2018; Morris et al., 2019). However, it has been shown that certain changes in the GNN architecture, such as the incorporation of high-order information (Morris et al., 2019; Maron et al., 2019) or the encoding of local or global structural features (Bouritsas et al., 2022; Feng et al., 2022) can improve the representational power of the resulting GNNs beyond that of classical MPGNNs. However, often the increase in expressivity comes at the cost of a significant increase in computational complexity.

**Structural and Positional Encodings.** Structural (SE) and Positional (PE) encodings endow MPGNNs with structural information that it cannot learn on its own, but which is crucial for downstream performance. Encoding approaches can capture either local or global information. Local PE endow nodes with information on its position within a local cluster or substructure, whereas global PE provide information on the nodes' global position in the graph. As such, PE are usually derived from distance notions. Examples of local PE include the distance of a node to the centroid of a cluster or community it is part of. Global PE often relate to spectral properties of the graph, such as the eigenvectors of the Graph Laplacian (Dwivedi et al., 2023) or random-walk based node similarities (Dwivedi et al., 2022a). Global PE are generally observed to be more effective than local PE. In contrast, SE encode structural similarity measures, either by endowing nodes with information about the subgraph they are part of or about the global topology of the graph. Notable examples of local SE are substructure counts (Bouritsas et al., 2022; Zhao et al., 2022), which are among the most popular encodings. Global SE often encode graph characteristics or summary statistics that an MPGNN is not able to learn on its own, e.g., its diameter, girth or the number of connected components (Loukas, 2019). In this work, we focus on Global PE and Local SE, as well as combinations of both types of encodings.

**Over-smoothing and Over-squashing.** MPGNNs may also suffer from over-squashing and over-smoothing effects. Over-squashing, first described by Alon & Yahav (2021) characterizes difficulties in leveraging information encoded in long-range connections, which is often crucial for downstream performance. Over-smoothing (Li et al., 2018) describes challenges in distinguishing node representations of nearby nodes, which occurs in deeper GNNs. While over-squashing is known to particular affect graph-level tasks, difficulties related to over-smoothing arise in particular in node-level tasks. Among the architectures considered here, GCN and GIN are prone to both effects, as they imple-

| Approach | Type | Encoding | Complexity | Geometric Information |
|---|---|---|---|---|
| LA (Dwivedi et al., 2023) | Global PE | Eigenvectors of Graph Laplacian. | $O(|V|^3)$ | spectral |
| RW (Dwivedi et al., 2022a) | Global PE | Landing probability of $k$-random walk. | $O(|V|d_{\max}^k)$ | $k$-hop commute time |
| SUB (Zhao et al., 2022) | Local SE | $k$-substructure counts. | $O(|V|^k)^*$ | motifs (size $k$) |
| LDP (Cai & Wang, 2018) | Local SE | Node degree distribution over neighborhood. | $O(|V|)$ | node degrees |
| **LCP (this paper)** | **Local SE** | **Curvature distribution over neighborhood.** | $O(|E|d_{\max}^3)^*$ | **motifs, 2-hop commute time** |

Table 1: Overview encoding approaches, where $d_{\max}$ denotes the highest node degree in the graph. ($*$: variants with lower complexity available)

ment sparse message-passing. In contrast, GAT improves the encoding of long-range dependencies using attention scores, which alleviates over-smoothing and over-squashing. A common approach for mitigating both effects in GCN, GIN and GAT is graph rewiring (Karhadkar et al., 2023; Topping et al., 2022; Nguyen et al., 2023; Fesser & Weber, 2023).

## 2.2 DISCRETE CURVATURE

Throughout this paper we utilize discrete Ricci cuvature, a central tool from Discrete Geometry that allows for characterizing (local) geometric properties of graphs. Discrete Ricci curvatures are defined via curvature analogies, mimicking classical properties of continuous Ricci curvature in the discrete setting. A number of such notions have been proposed in the literature; in this work, we mainly utilize a notion by Ollivier (Ollivier, 2009), which we introduce below. Others include Forman's Ricci curvature, which we introduce in appendix A.1.1.

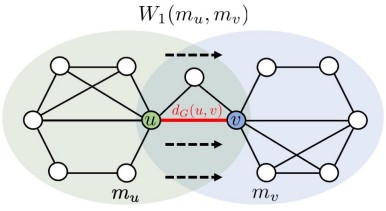

Figure 1: Computing ORC.

**Ollivier's Ricci Curvature.** Ollivier's notion of Ricci curvature derives from a connection of Ricci curvature and optimal transport. Suppose we endow the 1-hop neighborhoods of two neighboring vertices $u, v$, with uniform measures $m_i(z) := \frac{1}{\deg(i)}$, where $z$ is a neighbor of $i$ and $i \in \{u, v\}$. The transportation cost between the two neighborhoods along the edge $e = (u, v)$ can be measured by the Wasserstein-1 distance between the measures, i.e.,

$$W_1(m_u, m_v) = \inf_{m \in \Gamma(m_u, m_v)} \int_{(z,z') \in V \times V} d(z, z') m(z, z') \, dz \, dz' \tag{1}$$

where $\Gamma(m_u, m_v)$ denotes the set of all measures over $V \times V$ with marginals $m_u, m_v$. *Ollivier's Ricci curvature* (Ollivier, 2009) (short: ORC) is defined as

$$\kappa(u, v) := 1 - \frac{W_1(m_u, m_v)}{d_G(u, v)} \,, \tag{2}$$

where $d_G(u, v)$ denotes the distance between $u$ and $v$ in the graph $G$. The computation of ORC is shown schematically in Fig. 1.

**Graph Rewiring.** Previous applications of discrete Ricci curvature to GNNs include *rewiring*, i.e., approaches that add and remove edges to mitigate over-smoothing and over-squashing effects (Karhadkar et al., 2023; Topping et al., 2022; Nguyen et al., 2023; Fesser & Weber, 2023). It has been observed that long-range connections, which cause over-squashing effects, have low (negative) ORC (Topping et al., 2022), whereas oversmoothing is caused by edges of positive curvature in densely connected regions of the graph (Nguyen et al., 2023).

**Computational Aspects.** The computation of ORC can be expensive, as it requires solving an optimal transport problem for each edge in the graph, each of which is $O(d_{\max}^3)$ (via the Hungarian

algorithm). Faster approaches via Sinkhorn distances or combinatorial approximations of ORC exist (Tian et al., 2023), but can be much less accurate. In contrast, variants of Forman's curvature can be computed in as little as $O(d_{\max})$ per edge (see appendix A.1.1).

## 3 Encoding geometric structure with Structural Encodings

### 3.1 Local Curvature Profile

Curvature-based rewiring methods generally only affect the most extreme regions of the graph. They compute the curvature of every edge in the graph, but only add edges to the neighborhoods of the most negatively curved edges to resolve bottlenecks (Topping et al. (2022)). Similarly, they only remove the most positively curved edges to reduce over-smoothing (Nguyen et al. (2023), Fesser & Weber (2023)). Regions of the graph that have no particularly positively or negatively curved edges are not affected by the rewiring process, even though the curvature of the edges in these neighborhoods has also been computed. We believe that the curvature information of edges away from the extremes of the curvature distribution, which is not being used by curvature-based rewiring methods, can be beneficial to GNN performance.

We therefore propose to use curvature – more specifically the ORC – as a structural node encoding. Our *Local Curvature Profile* (LCP) adds summary statistics of the local curvature distribution to each node's features. Formally, for a given graph $G = (V, E)$ with vertex set $V$ and edge set $E$, we denote the multiset of the curvature of all incident edges of a node $v \in V$ by CMS (curvature multiset), i.e. $\text{CMS}(v) := \{\kappa(u, v) : (u, v) \in E\}$. We now define the Local Curvature Profile to consist of the following five summary statistics of the CMS:

$$\text{LCP}(v) := [\min(\text{CMS}(v)), \max(\text{CMS}(v)), \text{mean}(\text{CMS}(v)), \text{std}(\text{CMS}(v)), \text{median}(\text{CMS}(v))]$$

While our experiments in the main text are based on this notion of the LCP, other quantities from the CMS could be included. We provide results based on alternative notions of the LCP in appendix A.4. Note that computing the LCP as a preprocessing step requires us to calculate the curvature of each edge in $G$ exactly once. Our approach therefore has the same computational complexity as curvature-based rewiring methods, but uses curvature information everywhere in the graph.

### 3.2 Theoretical results

We will see below that encoding geometric characterization of substructures via LCP leads to an empirical benefit. Can we measure this advantage also with respect to the representational power of the GNN?

**Theorem 1.** *MPGNNs with LCP structural encodings are strictly more expressive than the 1-WL test and hence than MPGNNs without encodings.*

*Proof.* Seminal work by (Xu et al., 2018; Morris et al., 2019) has established that standard MPGNNs, such as GIN, are as expressive as the 1-WL test. It can be shown via a simple adaption of (Xu et al., 2018, Thm. 3) that adding LCP encodings does not decrease their expressivity, i.e., MPGNNs with LCP are at least as expressive as the 1-WL test. To establish strictly better expressivity, it is sufficient to identify a set of non-isomorphic graphs that cannot be distinguished by the 1-WL test, but that differ in

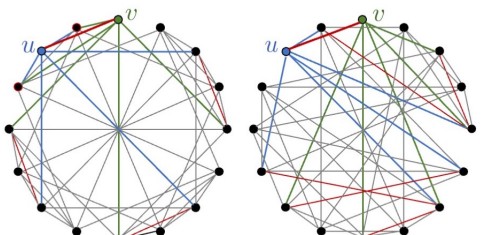

Figure 2: Illustration of optimal transport plans for computing the ORC of $(u, v)$ in the 4x4 Rooke (left) and Shrikhande (right) graphs. Edges along which a mass of $\frac{1}{deg(u)} = \frac{1}{deg(v)} = \frac{1}{6}$ is moved are marked in red. We see that $\kappa(u, v) = \frac{1}{3}$ in the Rooke and $\kappa(u, v) = 0$ in the Shrikhande graph.

their local curvature profiles. Consider the 4x4 Rooke and Shrikhande graphs, two non-isomorphic, strongly regular graphs. While nodes in both graphs have identical node degrees (e.g., could not be distinguished with classical MPGNNs), their 2-hop connectivities differ. This results in differences in the optimal transport plan of the neighborhood measures that is used to compute ORC (see

Fig. 2). As a result, the (local) ORC distribution across the two graphs varies, resulting in different local curvature profiles. □

**Remark 1.** It has been shown that encoding certain local substructure counts can allow MPGNNs to distinguish the 4x4 Rooke and Shrikhande graphs, establishing that such encodings lead to strictly more expressive MPGNNs (Bouritsas et al., 2022). However, this relies on a careful selection of the type of substructures to encode, a design choice that is expensive to fine-tune in practise, due to the combinatorial nature of the underlying optimization problem. In contrast, LCP implicitly characterizes substructure information via their impact on the local curvature distribution. This does not require fine-tuning of design choices. We note that spectral encodings cannot distinguish between the 4x4 Rooke and Shrikhande graphs, as their spectra are identical. LDP encodings cannot distinguish them either, as they have identical node degree distributions. Positional encodings based on random walks can distinguish them, but only stochastically.

We note that the relationship between the ORC curvature distribution and the WL hierarchy has been recently discussed in (Southern et al., 2023). They show that the 4x4 Rooke and Shrikhande graphs cannot be distinguished by the 2-WL or 3-WL test, indicating that ORC, and hence LCP encodings, can distinguish some graphs that cannot be distinguished with higher-order WL tests. However, the relationship between ORC and the WL hierachy remains an open question.

## 4 EXPERIMENTS

In this section, we experimentally demonstrate the effectiveness of the LCP as a structural encoding on a variety of tasks, including node and graph classification. In particular, we aim to answer the following questions:

Q1 Does LCP enhance the performance of GNNs on node- and graph-level tasks?

Q2 Are structural and positional encodings complementary? Do they encode different types of geometric information that can jointly enhance the performance of GNNs?

Q3 How does LCP compare to (curvature-based) rewiring in terms of accuracy?

We complement our main experiments with an investigation of other choices of curvature notions. In general, our experiments are meant to ensure a high level of fairness and comprehensiveness. Obtaining the best possible performance for each dataset presented is not our primary goal here.

### 4.1 EXPERIMENTAL SETUP

We treat the computation of encodings or rewiring as a preprocessing step, which is first applied to all graphs in the data sets considered: positional or structural encodings are concatenated to the node feature vectors, unless stated otherwise. We then train a GNN on a part of the rewired graphs and evaluate its performance on a withheld set of test graphs. As GNN architectures, we consider GCN (Kipf & Welling, 2017), GIN (Xu et al., 2018), and GAT(Veličković et al. (2018)). Settings and optimization hyperparameters are held constant across tasks and baseline models for all encoding and rewiring methods, so that hyperparameter tuning can be ruled out as a source of performance gain. We obtain the settings for the individual encoding types via hyperparameter tuning. For rewiring, we use the heuristics introduced by (Fesser & Weber, 2023). For all preprocessing methods and hyperparameter choices, we record the test set accuracy of the settings with the highest validation accuracy. As there is a certain stochasticity involved, especially when training GNNs, we accumulate experimental results across 100 random trials. We report the mean test accuracy, along with the $95\%$ confidence interval. Details on all data sets, including summary statistics, can be found in appendix A.10. Additional results on two of the long-range graph benchmark datasets introduced by (Dwivedi et al., 2022b) can also be found in appendix A.6.

### 4.2 RESULTS

#### 4.2.1 PERFORMANCE OF LCP (Q1)

Table 2 presents the results of our experiments for graph classification (Enzymes, Imdb, Mutag, and Proteins datasets) and node classification (Cora and Citeseer) with only the original node features,

i.e. no additional encodings (NO), Laplacian eigenvectors (LA) (Dwivedi et al., 2023), Random Walk transition probabilities (RW) (Dwivedi et al., 2022a), substructures (SUB) (Zhao et al., 2022), and our Local Curvature Profile (LCP). LCP outperforms all other encoding methods on all graph classification datasets. The improvement gained from using LCP is particularly impressive for GCN and GAT: the mean accuracy increases by between 10 (Enzymes) and almost 20 percent (Imdb) compared to using no encodings, and between 4 (Enzymes) and 14 percent compared to the second best encoding method.

On the node classification data sets in Table 2, LCP is still competitive, but the performance gains are generally much smaller, and other encoding methods occasionally outperform LCP. Additional experiments on other node classification datasets with similar results can be found in appendix A.7.

| MODEL | ENZYMES | IMDB | MUTAG | PROTEINS | CORA | CITE. |
|---|---|---|---|---|---|---|
| GCN (NO) | $25.4 \pm 1.3$ | $48.1 \pm 1.0$ | $62.7 \pm 2.1$ | $59.6 \pm 0.9$ | $86.6 \pm 0.8$ | $71.7 \pm 0.7$ |
| GCN (LA) | $26.5 \pm 1.1$ | $53.4 \pm 0.8$ | $70.8 \pm 1.7$ | $65.9 \pm 0.7$ | $88.0 \pm 0.9$ | $75.9 \pm 1.2$ |
| GCN (RW) | $29.7 \pm 2.5$ | $47.8 \pm 1.2$ | $67.0 \pm 3.2$ | $55.9 \pm 1.1$ | $87.6 \pm 1.1$ | $76.3 \pm 1.5$ |
| GCN (SUB) | $31.0 \pm 2.2$ | $51.2 \pm 1.0$ | $69.0 \pm 2.8$ | $61.1 \pm 0.8$ | $88.1 \pm 0.9$ | $76.9 \pm 1.1$ |
| GCN (LCP) | $35.4 \pm 2.6$ | $67.7 \pm 1.7$ | $79.0 \pm 2.9$ | $70.9 \pm 1.6$ | $88.9 \pm 1.0$ | $77.1 \pm 1.2$ |
| GIN (NO) | $29.7 \pm 1.1$ | $67.1 \pm 1.3$ | $67.5 \pm 2.7$ | $69.4 \pm 1.1$ | $76.3 \pm 0.6$ | $59.9 \pm 0.6$ |
| GIN (LA) | $26.6 \pm 1.9$ | $68.1 \pm 2.8$ | $74.0 \pm 1.4$ | $72.3 \pm 1.4$ | $80.1 \pm 0.7$ | $61.4 \pm 1.3$ |
| GIN (RW) | $27.7 \pm 1.4$ | $69.3 \pm 2.2$ | $76.0 \pm 2.7$ | $71.8 \pm 1.2$ | $78.1 \pm 1.0$ | $64.3 \pm 1.1$ |
| GIN (SUB) | $27.5 \pm 2.1$ | $68.2 \pm 2.1$ | $79.5 \pm 2.9$ | $71.5 \pm 1.1$ | $79.3 \pm 1.1$ | $62.1 \pm 1.3$ |
| GIN (LCP) | $32.7 \pm 1.6$ | $70.6 \pm 1.3$ | $82.1 \pm 3.8$ | $73.2 \pm 1.2$ | $79.8 \pm 1.1$ | $63.8 \pm 1.0$ |
| GAT (NO) | $22.5 \pm 1.7$ | $47.0 \pm 1.4$ | $68.5 \pm 2.7$ | $72.6 \pm 1.2$ | $83.4 \pm 0.8$ | $72.6 \pm 0.8$ |
| GAT (LA) | $23.2 \pm 1.3$ | $49.1 \pm 1.7$ | $71.0 \pm 2.6$ | $74.2 \pm 1.3$ | $84.7 \pm 1.0$ | $75.1 \pm 1.1$ |
| GAT (RW) | $23.4 \pm 1.7$ | $49.7 \pm 1.3$ | $70.8 \pm 2.4$ | $73.0 \pm 1.2$ | $83.8 \pm 0.9$ | $75.8 \pm 1.4$ |
| GAT (SUB) | $25.0 \pm 1.4$ | $50.9 \pm 1.3$ | $72.4 \pm 2.7$ | $75.7 \pm 1.5$ | $85.0 \pm 1.2$ | $76.3 \pm 1.2$ |
| GAT (LCP) | $34.5 \pm 2.0$ | $66.2 \pm 1.1$ | $82.0 \pm 1.9$ | $78.5 \pm 1.4$ | $85.7 \pm 1.1$ | $76.6 \pm 1.2$ |

Table 2: Graph (Enzymes, Imdb, Mutag, and Proteins) and node classification (Cora and Citeseer) accuracies of GCN, GIN, and GAT with positional, structural, or no encodings. Best results for each model highlighted in blue.

### 4.2.2 COMBINING STRUCTURAL AND POSITIONAL ENCODINGS (Q2)

To answer the question if and when structural and positional encodings are complimentary, we repeat the experiments from the previous subsection, only this time we combine one of the two structural encodings considered (SUB and LCP) with one of the positional encodings (LA and RW). The results are shown in Table 3. While we find that the best performing combination always includes LCP in all scenarios, the performance gains achieved depend on the model used. Using GCN, combining LCP with Laplacian eigenvectors or Random Walk transition probabilities improves performance on three of our six datasets, with Mutag showing the most significant gains (plus 7 percent). Using GIN, we see a performance improvement on five of our six datasets, with Proteins showing the largest gains (plus 3 percent). Finally, GAT shows performance gains on four of our six datasets, although those gains never exceed two 2 percent.

When comparing the accuracies attained by combining positional and structural encodings (Table 3) with the accuracies attained with only one positional or structural encoding (Table 2), we note that combinations generally result in better performance. However, we also note that the right choice of positional encoding seems to depend on the dataset: Random Walk transition probabilities lead to higher accuracy in 14 of 18 cases overall, Laplacian eigenvectors only in 4 cases. We believe that these differences stem from the different topologies of the graphs in our dataset, whose geometric properties may be captured better by certain encoding types than others (see also Table 2.1).

### 4.2.3 COMPARING STRUCTURAL ENCODING AND REWIRING (Q3)

In the last set of experiments, we compare the LCP, i.e. the use of curvature as a structural encoding, with curvature-based rewiring. We apply BORF Nguyen et al. (2023), an ORC-based rewiring strategy, to the data sets used so far. Table 4 shows the performance of our three model architectures on the rewired graphs without any positional encodings (NO) and with Laplacian eigenvectors (LA) or Random Walk transition probabilities (RW). Comparing the (NO) rows in Table 4 with the

| MODEL | ENZYMES | IMDB | MUTAG | PROTEINS | CORA | CITE. |
|---|---|---|---|---|---|---|
| GCN (LCP, LA) | $30.2 \pm 2.1$ | $60.3 \pm 1.0$ | $83.4 \pm 3.1$ | $70.1 \pm 1.6$ | $88.2 \pm 1.3$ | $76.7 \pm 1.4$ |
| GCN (LCP, RW) | $34.8 \pm 1.7$ | $62.6 \pm 1.2$ | $86.1 \pm 2.7$ | $69.7 \pm 1.5$ | $89.4 \pm 1.4$ | $77.5 \pm 1.3$ |
| GCN (SUB, LA) | $27.8 \pm 1.2$ | $48.4 \pm 1.0$ | $76.5 \pm 3.3$ | $65.8 \pm 1.3$ | $87.5 \pm 1.1$ | $76.2 \pm 1.3$ |
| GCN (SUB, RW) | $29.8 \pm 1.5$ | $54.3 \pm 1.4$ | $65.5 \pm 3.9$ | $59.1 \pm 1.3$ | $87.8 \pm 1.2$ | $74.4 \pm 1.5$ |
| GIN (LCP, LA) | $33.6 \pm 2.7$ | $70.8 \pm 1.1$ | $79.2 \pm 1.9$ | $75.7 \pm 1.2$ | $77.8 \pm 1.5$ | $63.2 \pm 1.2$ |
| GIN (LCP, RW) | $31.7 \pm 2.4$ | $72.1 \pm 1.7$ | $82.4 \pm 1.8$ | $76.2 \pm 1.4$ | $76.1 \pm 1.6$ | $66.3 \pm 1.1$ |
| GIN (SUB, LA) | $28.3 \pm 1.9$ | $68.3 \pm 1.4$ | $79.3 \pm 1.9$ | $73.4 \pm 1.1$ | $77.4 \pm 1.3$ | $64.1 \pm 1.2$ |
| GIN (SUB, RW) | $28.7 \pm 2.4$ | $69.9 \pm 1.1$ | $81.2 \pm 2.4$ | $71.9 \pm 1.2$ | $75.8 \pm 1.6$ | $62.3 \pm 1.4$ |
| GAT (LCP, LA) | $33.6 \pm 2.1$ | $64.3 \pm 1.2$ | $81.0 \pm 2.7$ | $78.7 \pm 1.6$ | $86.6 \pm 1.4$ | $77.2 \pm 1.3$ |
| GAT (LCP, RW) | $35.1 \pm 2.4$ | $65.2 \pm 1.8$ | $81.2 \pm 2.4$ | $79.4 \pm 1.7$ | $86.1 \pm 1.5$ | $77.4 \pm 1.5$ |
| GAT (SUB, LA) | $22.9 \pm 1.8$ | $50.4 \pm 1.6$ | $72.5 \pm 2.9$ | $76.0 \pm 1.6$ | $85.3 \pm 1.3$ | $76.9 \pm 1.6$ |
| GAT (SUB, RW) | $26.5 \pm 1.3$ | $48.0 \pm 1.5$ | $71.2 \pm 2.8$ | $75.8 \pm 1.7$ | $85.7 \pm 1.4$ | $77.1 \pm 1.4$ |

Table 3: Graph (Enzymes, Imdb, Mutag, and Proteins) and node classification (Cora and Citeseer) accuracies of GCN, GIN, and GAT with combinations of positional and structural encodings. Best results for each model highlighted in blue.

(LPC) rows in Table 2, we see that using the ORC to compute the LCP results in significantly higher accuracy on all data sets, compared to using it to rewire the graph. We believe that an intuitive explanation for these performance differences might be that rewiring uses a global curvature distribution, i.e. it compares the curvature values of all edges and then adds or removes edge at the extremes of the distribution. The LCP, on the other hand, is based on a local curvature distribution, so the LCP could be considered more faithful to the idea that the Ricci curvature is an inherently local notion.

As an extension, we also ask whether one should combine positional encodings with the LCP, or use them on the original graph before rewiring, to maintain some information of the original graph once rewiring has added and removed edges. Comparing the (LA) and (RW) rows in Table 4 with the (LCP, LA) and (LCP, RW) rows in Table 3, we see that the LCP-based variant outperforms the rewiring-based one on all graph classification data sets. Combining rewiring and positional encodings attains the best performance in only two cases on the node classification data sets.

| MODEL | ENZYMES | IMDB | MUTAG | PROTEINS | CORA | CITE. |
|---|---|---|---|---|---|---|
| GCN (NO) | $26.0 \pm 1.2$ | $48.6 \pm 0.9$ | $68.2 \pm 2.4$ | $61.2 \pm 0.9$ | $87.9 \pm 0.7$ | $73.4 \pm 0.6$ |
| GCN (LA) | $26.3 \pm 1.7$ | $53.7 \pm 1.2$ | $75.5 \pm 2.8$ | $64.4 \pm 1.2$ | $86.1 \pm 1.0$ | $74.7 \pm 0.8$ |
| GCN (RW) | $24.0 \pm 1.8$ | $49.4 \pm 1.1$ | $74.0 \pm 2.8$ | $61.9 \pm 1.1$ | $88.2 \pm 0.9$ | $75.7 \pm 0.8$ |
| GIN (NO) | $31.9 \pm 1.2$ | $67.7 \pm 1.5$ | $75.4 \pm 2.8$ | $72.3 \pm 1.2$ | $78.4 \pm 0.8$ | $63.1 \pm 0.7$ |
| GIN (LA) | $28.1 \pm 1.6$ | $70.2 \pm 2.1$ | $78.3 \pm 2.1$ | $75.2 \pm 1.3$ | $77.3 \pm 1.1$ | $64.3 \pm 1.0$ |
| GIN (RW) | $28.4 \pm 1.7$ | $71.7 \pm 2.4$ | $78.0 \pm 1.6$ | $74.1 \pm 1.3$ | $78.6 \pm 1.3$ | $64.2 \pm 1.1$ |
| GAT (NO) | $21.7 \pm 1.5$ | $47.1 \pm 1.6$ | $72.2 \pm 2.1$ | $73.6 \pm 1.4$ | $83.8 \pm 1.1$ | $73.4 \pm 0.9$ |
| GAT (LA) | $25.3 \pm 1.5$ | $52.9 \pm 1.2$ | $74.4 \pm 1.8$ | $74.9 \pm 1.5$ | $85.3 \pm 1.5$ | $75.4 \pm 1.2$ |
| GAT (RW) | $22.0 \pm 1.7$ | $52.1 \pm 0.9$ | $68.5 \pm 2.0$ | $74.1 \pm 1.7$ | $84.2 \pm 1.2$ | $76.1 \pm 1.6$ |

Table 4: Graph (Enzymes, Imdb, Mutag, and Proteins) and node classification (Cora and Citeseer) accuracies of GCN, GIN, and GAT with positional, structural, or no encodings, on graphs rewired using BORF.

**LCP and GNN depth.** Rewiring strategies are often used to mitigate over-smoothing, for example by removing edges in particularly dense neighborhoods of the graph (Fesser & Weber (2023)), which in turn allows for the use of deeper GNNs. As Figure 4.3 shows, using the LCP as a structural encoding can help in this regard as well: the average accuracy of a GCN with LCP structural encodings (y-axis) does not decrease faster with the number of layers (x-axis) than the average accuracy of a GCN on a rewired graph. Both ways of using ORC lose between 4 and 5 percent as we move from 5 GCN layers to 10.

## 4.3    DIFFERENT NOTIONS OF CURVATURE

So far, we have implemented LCP with Ollivier-Ricci curvature in all experiments. However, other notions of curvature have been considered in Graph Machine Learning applications, including in rewiring. (Fesser & Weber (2023)) use the Augmented Forman-Ricci curvature (AFRC), which enriches the Forman-Ricci curvature (FRC) by considering 3-cycles (AFRC-3) or 3- and 4-cycles

(AFRC-4). For details on FRC and AFRC, see appendix A.1.1. AFRC-3 and AFRC-4 in particular are cheaper to compute than the ORC and have been show to result in competitive rewiring methods (Fesser & Weber (2023)). As such, one might ask if we could not also use one of these curvature notions to compute the LCP.

| LCP Curvature | ENZYMES | IMDB | MUTAG | PROTEINS |
|---|---|---|---|---|
| FRC | $27.4 \pm 1.1$ | $69.6 \pm 1.1$ | $72.0 \pm 2.1$ | $64.1 \pm 1.3$ |
| AFRC-3 | $28.0 \pm 1.8$ | $50.7 \pm 1.1$ | $72.3 \pm 1.8$ | $62.6 \pm 1.5$ |
| AFRC-4 | $29.2 \pm 2.4$ | $54.4 \pm 1.6$ | $74.5 \pm 3.1$ | $64.2 \pm 1.5$ |
| ORC | $35.4 \pm 2.6$ | $67.7 \pm 1.7$ | $79.0 \pm 2.9$ | $70.9 \pm 1.6$ |

Table 5: Graph classification accuracies of GCN with LCP structural encodings using FRC, AFRC-3, AFRC-4, and ORC. Best results highlighted in blue.

We train the same GCN used in the previous sections with the LCP structural encoding on our graph classification data sets. Table 5 shows the mean accuracies attained using different curvatures to compute the LCP. We note that all four curvatures improve upon the baseline (no structural encoding), and that generally, performance increases as we move from the FRC to its augmentations and finally to the ORC. This might not come as a surprise, since the FRC and its augmentations can be thought of as low-level approximations of the ORC (Jost & Münch (2021)). Our results therefore seem to suggest that when we use curvature as a structural encoding, GNNs are generally able to extract the additional information contained in the ORC. This does not seem to be true for curvature-based rewiring methods, where we can use AFRC-3 or AFRC-4 without significant performance drops (Nguyen et al. (2023), Fesser & Weber (2023)).

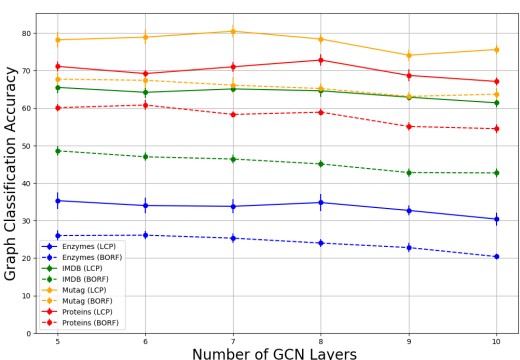

Figure 3: Graph classification accuracy with increasing number of GCN layers. Dashed lines show accuracies using BORF, normal lines accuracy using the LCP.

Our experimental results also suggest that the classical Forman curvature is surprisingly effective as a faster substitute for ORC. The choice of which higher-order structures to encode (i.e., AFRC-3 vs. AFRC-4) impacts the ability of LCP to characterize substructures with higher-order cycles. Hence, depending on the graph topology, the choice of FRC or AFRC over ORC will affect performance. For example, in the case of Imdb, using FRC seems to work well, which we attribute to the special topology of the graphs in the Imdb dataset (see appendix A.9 for more details).

## 5 DISCUSSION AND LIMITATIONS

In this paper we have introduced Local Curvature Profiles (LCP) as a novel structural encoding, which endows nodes with a geometric characterization of their neighborhoods with respect to discrete curvature. Our experiments have demonstrated that LCP outperforms existing encoding approaches and that further performance increases can be achieved by combining LCP with global positional encodings. In comparison with curvature-based rewiring, a previously proposed technique for utilizing discrete curvature in GNNs, LCP achieves superior performance.

While we establish that LCP improves the expressivity of MPGNNs, the relationship between LCP and the $k$-WL hierarchy remains open. Our experiments have revealed differences in the effectiveness of encodings, including LCP, between GNN architectures. An important direction for future work is a more detailed analysis of the influence of model design (type of layer, depths, choice of hyperparameter) with respect to the graph learning task (node- vs. graph-level) and the topology of the input graph(s). In addition, while we have investigated the choice of curvature notion from a computational complexity perspective, further research on the suitability of each notion for different tasks and graph topologies is needed.

ACKNOWLEDGEMENTS

MW was partially supported by NSF award 2112085.

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

# A  APPENDIX

CONTENTS

### A.1 OTHER CURVATURE NOTIONS

#### A.1.1 FORMAN'S RICCI CURVATURE

Ricci curvature is a classical tool in Differential Geometry, which establishes a connection between the geometry of the manifold and local volume growth. Discrete notions of curvature have been proposed via *curvature analogies*, i.e., notions that maintain classical relationships with other geometric characteristics. Forman (Forman, 2003) introduced a notion of curvature on CW complexes, which allows for a discretization of a crucial relationship between Ricci curvature and Laplacians, the Bochner-Weizenböck equation. In the case of a simple, undirect, and unweighted graph $G = (V, E)$, the Forman-Ricci curvature of an edge $e = (u, v) \in E$ can be expressed as

$$\mathcal{FR}(u, v) = 4 - \deg(u) - \deg(v)$$

#### A.1.2 AUGMENTED FORMAN-RICCI CURVATURE.

The edge-level version of Forman's curvature above also allows for evaluating curvature contributions of higher-order structures. Specifically, (Fesser & Weber (2023)) consider notions that evaluate higher-order information encoded in cycles of order $\leq k$ (denoted as $\mathcal{AF}_k$), focusing on the cases $k = 3$ and $k = 4$:

$$\mathcal{AF}_3(u, v) = 4 - \deg(u) - \deg(v) + 3\triangle(u, v)$$
$$\mathcal{AF}_4(u, v) = 4 - \deg(u) - \deg(v) + 3\triangle(u, v) + 2\square(u, v) \ ,$$

where $\triangle(u, v)$ and $\square(u, v)$ denote the number of triangles and quadrangles containing the edge $(u, v)$. The derivation of those notions follows directly from (Forman, 2003) and can be found, e.g., in (Tian et al., 2023).

#### A.1.3 COMBINATORIAL APPROXIMATION OF ORC

Below, we will utilize combinatorial upper and lower bounds on ORC for a more efficiently computable approximation of local curvature profiles. The bounds were first proven by (Jost & Liu, 2014), we recall the result below for the convenience of the reader. A version for weighted graphs can be found in (Tian et al., 2023).

**Theorem 2.** *We denote with $\#(u, v)$ the number of triangles that include the edge $(u, v)$ and $d_v$ the degree of $v$. Then the following bounds hold:*

$$-\left(1 - \frac{1}{d_v} - \frac{1}{d_u} - \frac{\#(u,v)}{d_u \wedge d_v}\right)_+ - \left(1 - \frac{1}{d_v} - \frac{1}{d_u} - \frac{\#(u,v)}{d_u \vee d_v}\right)_+ + \frac{\#(u,v)}{d_u \vee d_v} \leq \kappa(u, v) \leq \frac{\#(u,v)}{d_u \vee d_v}$$

*Here, we have used the shorthand $a \wedge b := \min\{a, b\}$ and $a \vee b := \max\{a, b\}$.*

#### A.1.4 COMPUTATIONAL COMPLEXITY

As discussed in the main text, the computation of the ORC underlying the LCP complexity has a complexity of $O(|E|d_{\max}^3)$. In contrast, the combinatorial bounds in the previous section require only the computation of node degree and triangle counts, which results in a reduction of the complexity to $O(|E|d_{\max})$. Classical Forman curvature ($\mathcal{FR}(\cdot, \cdot)$) can be computed in $O(|V|)$, but the resulting geometric characterization is often less informative. The more informative augmented notions have complexities $O(|E|d_{\max})$ ($\mathcal{AF}_3$) and $O(|E|d_{\max}^2)$ ($\mathcal{AF}_4$).

### A.2 BEST HYPERPARAMETER CHOICES

Our hyperparameter choices for structural and positional encodings are largely based on the hyperparameters reported in (Dwivedi et al. (2023), Dwivedi et al. (2022a), Zhao et al. (2022)). We used their values as starting values for a grid search, and found a random walk length of 16 to work best for RWPE, 8 eigenvectors to work best for LAPE, and a walk length of 10 to work best for SUB, with minimal differences across graph classification datasets. As mentioned in the main text, we did not use hyperparameter tuning for BORF, both rather used the heuristics proposed in (Fesser & Weber (2023)).

A.3 MODEL ARCHITECTURES

**Node classification.** We use a GNN with 3 layers and hidden dimension 128. We further use a dropout probability of 0.5, and a ReLU activation. We use this architecture for all node classification datasets in the paper.

**Graph classification.** We use a GNN with 4 layers and hidden dimension 64. We further use a dropout probability of 0.5, and a ReLU activation. We use this architecture for all graph classification datasets in the paper.

Unless explicitly stated otherwise, we train all models until we observe no improvements in the validation accuracy for 100 epochs using the Adam optimizer with learning rate 1e-3 and a batch size of 16. We use a train/val/test split of 50/25/25.

A.4 ALTERNATIVE DEFINITIONS OF THE LCP

In this subsection, we investigate several alternative definitionso of the LCP. As before, let $\text{CMS}(v) := \{\kappa(u, v) : (u, v) \in E\}$ denote the curvature multiset. Then we define the Local Curvature Profile using extreme values as

$$\text{LCP}(v) := [(\text{CMS}(v))_1, (\text{CMS}(v))_2, (\text{CMS}(v))_3, (\text{CMS}(v))_{n-1}, (\text{CMS}(v))_n]$$

where $n := \deg(v)$ and $(\text{CMS}(v))_1 \leq (\text{CMS}(v))_1 \leq ... \leq (\text{CMS}(v))_n$. Similarly, we can define the LCP using the minimum and maximum of the CMS only, i.e.

$$\text{LCP}(v) := [\min(\text{CMS}(v)), \max(\text{CMS}(v))]$$

or use the combinatorial upper and lower bounds for the ORC introduced earlier and then define the LCP using the minimum of the lower bounds and the maximum of the upper bounds. Formally, we let $\text{CMS}_u(v) := \{\kappa_u(u, v) : (u, v) \in E\}$ and $\text{CMS}_l(v) := \{\kappa_l(u, v) : (u, v) \in E\}$, where $\kappa_u(u, v)$ and $\kappa_l(u, v)$ are the combinatorial upper (lower) bounds of $\kappa(u, v)$. The LCP is then given by

$$\text{LCP}(v) := [\min(\text{CMS}_l(v)), \max(\text{CMS}_u(v))]$$

We present the graph classification results attained using these alternative definitions of the LCP in Table 6.

| LCP | ENZYMES | IMDB | MUTAG | PROTEINS |
|---|---|---|---|---|
| All Summary Statistics | $35.4 \pm 2.6$ | $67.7 \pm 1.7$ | $79.0 \pm 2.9$ | $70.9 \pm 1.6$ |
| Mean, Med, and Std | $30.2 \pm 2.5$ | $64.8 \pm 1.3$ | $76.4 \pm 3.0$ | $69.7 \pm 1.1$ |
| Min and Max | $33.5 \pm 1.9$ | $66.6 \pm 1.4$ | $78.5 \pm 2.4$ | $72.1 \pm 1.1$ |
| Min only | $29.0 \pm 1.7$ | $63.1 \pm 1.3$ | $66.5 \pm 2.3$ | $66.8 \pm 1.4$ |
| Max only | $29.8 \pm 2.1$ | $63.4 \pm 1.5$ | $75.4 \pm 2.6$ | $66.1 \pm 1.2$ |
| Extreme Values | $36.2 \pm 1.9$ | $65.6 \pm 2.1$ | $80.1 \pm 2.4$ | $70.4 \pm 1.5$ |
| Combinatorial Approx. | $32.7 \pm 1.7$ | $63.9 \pm 1.3$ | $84.2 \pm 2.1$ | $70.9 \pm 1.3$ |

Table 6: Graph classification accuracies of GCN with LCP structural encodings using summary statistics (top) and using the most extreme values (bottom).

A.5 COMPARISON WITH CURVATURE GRAPH NETWORK

**Attention weights vs. Curvature.** We remark on a previous work that utilizes discrete curvature in the design of MPGNN architectures, aside from the rewiring techniques discussed earlier. Curvature Graph Networks (Ye et al. (2020)) proposes to weight messages during the updating of the node representations by a function of the curvature of the corresponding edge. Formally, their version of the previously introduced update is given by

$$\mathbf{x}_v^{k+1} = \phi_k \left( \bigoplus_{p \in \tilde{\mathcal{N}}_v} \tau_{(v,p)}^k \mathbf{W}^k \mathbf{x}_p^k \right) ,$$

where $\mathbf{W}^k$ is a learned weight matrix and $\tau^k_{(v,p)}$ is a function of the ORC of the edge $(v, p)$, which is learned using an MLP. We note that this is analogous to the weighting of messages in GAT (Veličković et al. (2018)), where self attention plays the role of $\tau^k_{(v,p)}$. In fact, Curvature Graph Network attains similar performance to GAT, and even outperforms it on some node classification tasks. However, we attain even better performance at the same computational cost using the LCP (Table 7).

|  | CORA | CITESEER |
|---|---|---|
| CurvGN-1 | $83.1 \pm 0.8$ | $71.7 \pm 1.0$ |
| CurvGN-n | $83.2 \pm 0.9$ | $72.4 \pm 0.9$ |
| GCN (LCP) | $88.9 \pm 1.0$ | $77.1 \pm 1.2$ |
| GAT (LCP) | $85.7 \pm 1.1$ | $76.6 \pm 1.2$ |

Table 7: Node classification accuracy of curvature graph network vs. GCN and GAT with LCP structural encodings.

### A.6 RESULTS ON LONG-RANGE GRAPH BENCHMARK DATASETS

| MODEL | PEPTIDES-FUNC | PEPTIDES-STRUCT |
|---|---|---|
| GCN (NO) | $40.7 \pm 2.0$ | $0.379 \pm 0.013$ |
| GCN (LA) | $43.5 \pm 1.8$ | $0.356 \pm 0.014$ |
| GCN (RW) | $43.2 \pm 2.1$ | $0.354 \pm 0.019$ |
| GCN (SUB) | $42.6 \pm 2.0$ | $0.360 \pm 0.016$ |
| GCN (LCP) | $44.4 \pm 2.2$ | $0.352 \pm 0.017$ |
| GIN (NO) | $46.2 \pm 2.2$ | $0.387 \pm 0.023$ |
| GIN (LA) | $48.8 \pm 1.6$ | $0.364 \pm 0.021$ |
| GIN (RW) | $48.0 \pm 2.1$ | $0.368 \pm 0.023$ |
| GIN (SUB) | $47.3 \pm 2.4$ | $0.375 \pm 0.020$ |
| GIN (LCP) | $49.6 \pm 2.2$ | $0.361 \pm 0.022$ |

Table 8: Mean classification accuracy (Peptides-func) and mean absolute error (Peptides-struct) of GCN and GIN with positional, structural, or no encodings. Best results for each model highlighted in blue. Note that for Peptides-struct, lower is better.

| MODEL | PEPTIDES-FUNC | PEPTIDES-STRUCT |
|---|---|---|
| GCN (NO) | $43.8 \pm 2.6$ | $0.365 \pm 0.018$ |
| GCN (LA) | $45.2 \pm 2.3$ | $0.348 \pm 0.021$ |
| GCN (RW) | $44.5 \pm 2.2$ | $0.341 \pm 0.022$ |
| GIN (NO) | $49.3 \pm 1.8$ | $0.378 \pm 0.025$ |
| GIN (LA) | $50.1 \pm 2.1$ | $0.352 \pm 0.022$ |
| GIN (RW) | $50.4 \pm 2.5$ | $0.350 \pm 0.024$ |

Table 9: Mean classification accuracy (Peptides-func) and mean absolute error (Peptides-struct) of GCN and GIN with positional encodings on graphs rewired using BORF. Note that for Peptides-struct, lower is better.

## A.7 RESULTS ON OTHER NODE CLASSIFICATION DATASETS

| MODEL | CORNELL | TEXAS | WISCONSIN | AMAZON | MINESWEEPER | TOLOKERS |
|---|---|---|---|---|---|---|
| GCN (NO) | $46.8 \pm 2.6$ | $44.3 \pm 2.5$ | $43.8 \pm 0.7$ | $46.6 \pm 0.4$ | $80.5 \pm 0.4$ | $79.1 \pm 0.5$ |
| GCN (LA) | $49.4 \pm 2.1$ | $50.5 \pm 2.1$ | $47.3 \pm 1.1$ | $47.2 \pm 0.3$ | $80.5 \pm 0.3$ | $79.2 \pm 0.4$ |
| GCN (RW) | $48.3 \pm 2.4$ | $49.2 \pm 2.3$ | $47.1 \pm 1.3$ | $47.1 \pm 0.4$ | $81.6 \pm 0.9$ | TIMEOUT |
| GCN (SUB) | $49.6 \pm 2.1$ | $52.3 \pm 2.4$ | $46.8 \pm 1.2$ | $47.3 \pm 0.4$ | $81.8 \pm 1.0$ | $78.7 \pm 0.8$ |
| GCN (LCP) | $50.4 \pm 2.5$ | $56.8 \pm 2.6$ | $47.4 \pm 1.4$ | $47.5 \pm 0.3$ | $82.7 \pm 0.8$ | $80.6 \pm 0.7^*$ |
| GIN (NO) | $36.7 \pm 2.3$ | $54.1 \pm 3.0$ | $48.6 \pm 2.2$ | $47.5 \pm 0.4$ | $78.2 \pm 0.4$ | $78.6 \pm 0.2$ |
| GIN (LA) | $51.3 \pm 2.0$ | $67.8 \pm 3.3$ | $57.8 \pm 2.1$ | $47.7 \pm 0.4$ | $79.6 \pm 1.1$ | $78.3 \pm 0.6$ |
| GIN (RW) | $49.6 \pm 2.2$ | $62.2 \pm 3.5$ | $53.1 \pm 2.4$ | $47.0 \pm 0.3$ | $79.2 \pm 0.5$ | TIMEOUT |
| GIN (SUB) | $47.8 \pm 1.9$ | $60.5 \pm 2.8$ | $54.0 \pm 2.3$ | $47.4 \pm 1.0$ | $78.9 \pm 1.2$ | $78.2 \pm 0.9$ |
| GIN (LCP) | $50.5 \pm 2.6$ | $63.6 \pm 3.2$ | $53.8 \pm 2.4$ | $47.2 \pm 0.7$ | $80.1 \pm 0.9$ | $79.2 \pm 0.6^*$ |

Table 10: Node classification accuracies of GCN and GIN with positional, structural, or no encodings. Best results for each model highlighted in blue. *The LCP on the tolokers dataset was computed using the combinatorial approximations presented earlier. Computing the actual ORC takes longer than 60 minutes, which we consider a timeout.

## A.8 RESULTS ON ZINC DATASET

| MODEL | ZINC |
|---|---|
| GCN (No) | $0.397 \pm 0.011$ |
| GCN (LA) | $0.376 \pm 0.012$ |
| GCN (RW) | $0.371 \pm 0.017$ |
| GCN (SUB) | $0.375 \pm 0.016$ |
| GCN (LCP) | $0.363 \pm 0.017$ |
| GIN (NO) | $0.546 \pm 0.051$ |
| GIN (LA) | $0.522 \pm 0.058$ |
| GIN (RW) | $0.514 \pm 0.067$ |
| GIN (SUB) | $0.511 \pm 0.062$ |
| GIN (LCP) | $0.502 \pm 0.065$ |
| GAT (NO) | $0.404 \pm 0.007$ |
| GAT (LA) | $0.388 \pm 0.011$ |
| GAT (RW) | $0.382 \pm 0.014$ |
| GAT (SUB) | $0.379 \pm 0.012$ |
| GAT (LCP) | $0.372 \pm 0.015$ |

Table 11: MAE of GCN, GIN, and GAT with positional, structural, or positional and structural encodings. Best results for each model highlighted in blue.

## A.9 ADDITIONAL FIGURES

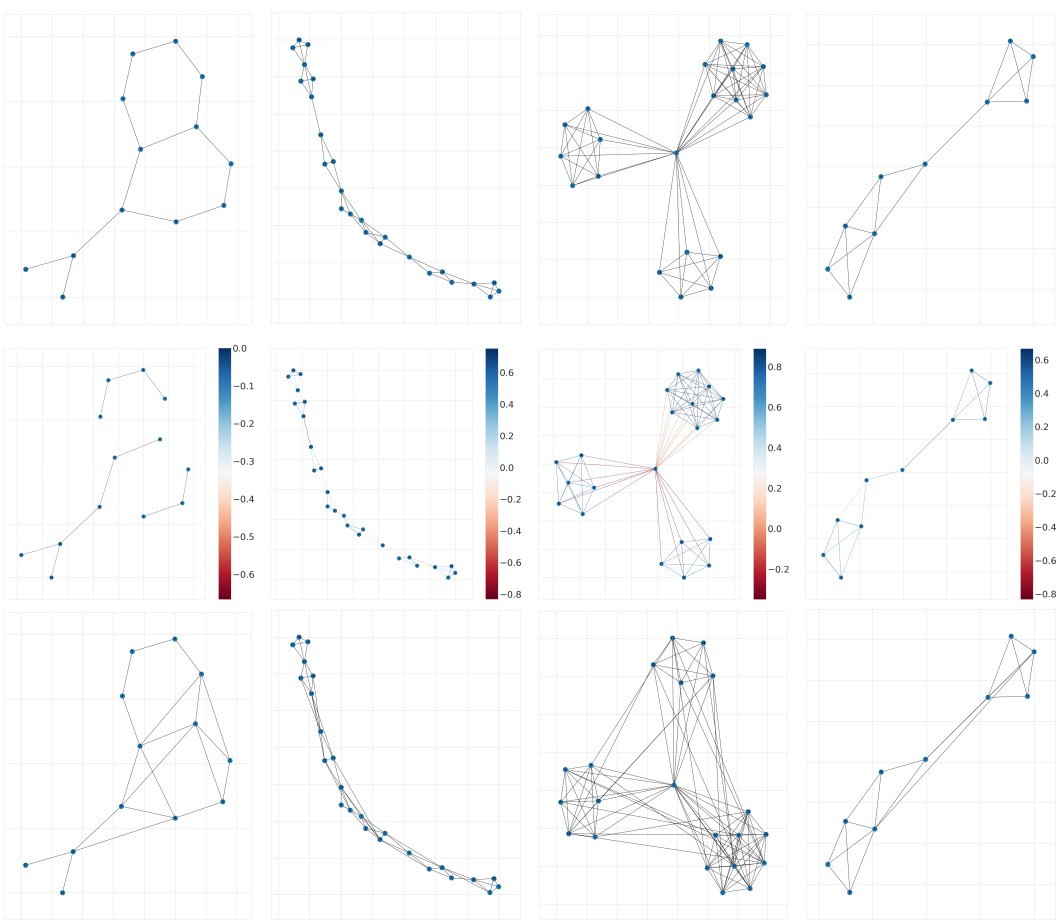

Figure 4: Example networks from the mutag, enzymes, imdb, and proteins datasets, which we use for graph classification (top row). The middle row shows the same example networks with their edges colored according to their ORC values. We also depict the adjusted graphs after rewiring using BORF (bottom row).

## A.10 STATISTICS FOR DATASETS IN THE MAIN TEXT

### A.10.1 GENERAL STATISTICS FOR NODE CLASSIFICATION DATASETS

|  | CORA | CITESEER |
|---|---|---|
| #NODES | 2485 | 2120 |
| #EDGES | 5069 | 3679 |
| # FEATURES | 1433 | 3703 |
| #CLASSES | 7 | 6 |
| DIRECTED | FALSE | FALSE |

Table 12: Statistics of node classification datasets.

### A.10.2 Curvature distributions for node classification datasets

| DATASET | MIN. | MAX. | MEAN | STD |
|---|---|---|---|---|
| CORA | $-0.898$ | 1.0 | 0.139 | 0.346 |
| CITESEER | $-0.861$ | 1.0 | 0.029 | 0.402 |

Table 13: Curvature (ORC) statistics of node classification datasets.

### A.10.3 General statistics for graph classification datasets

| | ENZYMES | IMDB | MUTAG | PROTEINS |
|---|---|---|---|---|
| #GRAPHS | 600 | 1000 | 188 | 1113 |
| #NODES | 2-126 | 12-136 | 10-28 | 4-620 |
| #EDGES | 2-298 | 52-2498 | 20-66 | 10-2098 |
| AVG #NODES | 32.63 | 19.77 | 17.93 | 39.06 |
| AVG #EDGES | 124.27 | 193.062 | 39.58 | 145.63 |
| #CLASSES | 6 | 2 | 2 | 2 |
| DIRECTED | FALSE | FALSE | FALSE | FALSE |

Table 14: Statistics of graph classification datasets.

### A.10.4 Curvature distributions for graph classification datasets

| DATASET | MIN. | MAX. | MEAN | STD |
|---|---|---|---|---|
| ENZYMES | $-0.382$ | 0.614 | 0.157 | 0.230 |
| IMDB | 0.007 | 0.606 | 0.394 | 0.223 |
| MUTAG | $-0.334$ | 0.344 | $-0.067$ | 0.218 |
| PROTEINS | $-0.335$ | 0.624 | 0.185 | 0.228 |

Table 15: Curvature (ORC) statistics of graph classification datasets.

**Datasets.** We conduct our node classification experiments on the publicly available CORA and CITESEER Yang et al. (2016) datasets, and our graph classification experiments on the ENZYMES, IMDB, MUTAG and PROTEINS datasets from the TUDataset collection Morris et al. (2020).

### A.11 Hardware specifications and libraries

We implemented all experiments in this paper in Python using PyTorch, Numpy PyTorch Geometric, and Python Optimal Transport. We created the figures in the main text using inkscape.

Our experiments were conducted on a local server with the specifications presented in the following table.

| COMPONENTS | SPECIFICATIONS |
|---|---|
| ARCHITECTURE | X86_64 |
| OS | UBUNTU 20.04.5 LTS x86_64 |
| CPU | AMD EPYC 7742 64-CORE |
| GPU | NVIDIA A100 TENSOR CORE |
| RAM | 40GB |

Table 16:

