# OpenReview forum: "Effective Structural Encodings via Local Curvature Profiles"
_ICLR.cc/2024/Conference — ICLR 2024 poster_

### Official Review · Reviewer_pewF · 2023-10-28

**Soundness:** 2 fair
**Presentation:** 2 fair
**Contribution:** 2 fair
**Rating:** 3
**Confidence:** 2

**Summary:**

The paper explores the efficacy of various structural encodings, along with their integration with global positional encodings, to enhance the performance of Graph Neural Networks (GNNs) in downstream tasks. It introduces a novel structural encoding known as LCP, derived from discrete Ricci curvature, which demonstrates superior performance compared to existing encoding methods.

**Strengths:**

1. The article introduces LCP encoding which presents how curvature information may affect GNN performance, contributing to a better understanding of the research context and significance
2. The article conducts comprehensive experiments on various datasets; however, it could benefit from additional experiments to investigate the underlying reasons why LCP is effective

**Weaknesses:**

1. The article lacks explanations for some crucial implementation steps, making it confusing to read. I suggest the author improve the presentation and logic of the article to enhance clarity (see question 1 and 2)
2. Please provide definitions for variable names, such as 'd_max.' Currently, there are many variables in the article that are not explained, which can be challenging for newcomers to understand
3. In spite of keeping settings and optimization hyperparameters consistent among different settings, the authors should still provide the corresponding parameter configurations. This would aid in experiment reproducibility and, as a result, make the results more robust.
4. In Section 3.1, the authors mention, 'We believe that the curvature information of edges away from the extremes of the curvature distribution, which is not being used by curvature-based rewiring methods, can be beneficial to GNN performance.' I consider this assertion somewhat speculative, and I did not find any subsequent experiments that substantiate this claim. Would it be possible to include relevant ablation experiments to support this hypothesis?
5. In Section 3.1, the authors define LCP as 'five summary statistics of the CMS.' I would appreciate a more detailed motivation for this particular definition. Additionally, it would be beneficial to include relevant ablation experiments that showcase the impact of removing specific summary statistics to demonstrate their significance in influencing the final results.

**Questions:**

1. Could the authors please provide a detailed explanation of the specific approach referred to as 'no encodings (NO)'?
2. Could the authors please elaborate on how the combination of LCP encoding and position encoding is implemented in Section 4.2.2? I couldn't find any details regarding the actual implementation.

---

> ### Author Response · Authors · 2023-11-15
>
> We would like to thank the reviewer for the detailed feedback.
>
> > Could the authors please provide a detailed explanation of the specific approach referred to as 'no encodings (NO)'?
>
> By ‘no encodings (NO)’, we refer to the setting where only the original node features are available to the GNN, no additional positional or structural encodings were added. We have added a short clarification of this in section 4.2.
>
> > Could the authors please elaborate on how the combination of LCP encoding and position encoding is implemented in Section 4.2.2? I couldn't find any details regarding the actual implementation.
>
> For the experiments in the main text, we concatenated the positional or structural encodings to the node feature vectors in the original datasets as a preprocessing step. When using two different encodings together, we concatenated both to the original feature vector. We have added a clarification on this in section 4.1. We also ran ablations on some of the datasets considered where instead of only concatenating encodings and thereby increasing the dimension of the node features, we linearly projected the concatenated node features down to the original node feature dimension (following [1]). This had no clear effects on performance, however.
>
> > Please provide definitions for variable names, such as 'd_max.' Currently, there are many variables in the article that are not explained, which can be challenging for newcomers to understand.
>
> We thank the reviewer for pointing this out. We have added explanations for all previously unintroduced variables.
>
> > In spite of keeping settings and optimization hyperparameters consistent among different settings, the authors should still provide the corresponding parameter configurations. This would aid in experiment reproducibility and, as a result, make the results more robust.
>
> We have extended appendix section A.3 and included relevant hyperparameter choices to make our results reproducible without considering other papers.
>
> > In Section 3.1, the authors mention, 'We believe that the curvature information of edges away from the extremes of the curvature distribution, which is not being used by curvature-based rewiring methods, can be beneficial to GNN performance.' I consider this assertion somewhat speculative, and I did not find any subsequent experiments that substantiate this claim. Would it be possible to include relevant ablation experiments to support this hypothesis?
>
> We consider the experiments in section 4.2 to be very strong evidence for the idea that the curvature information of edges away from the extremes of the curvature distribution can be useful for GNN performance. Using the LCP clearly outperforms methods that use the ORC to rewire the graph on all datasets.
>
> > In Section 3.1, the authors define LCP as 'five summary statistics of the CMS.' I would appreciate a more detailed motivation for this particular definition. Additionally, it would be beneficial to include relevant ablation experiments that showcase the impact of removing specific summary statistics to demonstrate their significance in influencing the final results.
>
> Our use of these summary statistics is motivated by their use in the Local Degree Profile (LDP) [2]. As our ablations in Appendix A.4 show, min and max seem to be the most useful summary statistics.
>
> [1] Rampášek, Ladislav, Michael Galkin, Vijay Prakash Dwivedi, Anh Tuan Luu, Guy Wolf, and Dominique Beaini. "Recipe for a general, powerful, scalable graph transformer." Advances in Neural Information Processing Systems 35 (2022): 14501-14515.
>
> [2] Cai, Chen, and Yusu Wang. "A simple yet effective baseline for non-attributed graph classification." arXiv preprint arXiv:1811.03508 (2018).

---

### Official Review · Reviewer_GHx9 · 2023-11-01

**Soundness:** 3 good
**Presentation:** 3 good
**Contribution:** 2 fair
**Rating:** 6
**Confidence:** 4

**Summary:**

The paper proposes to use local curvature profile (LCP) for structural encoding in graph neural networks. Several notions of local curvatures are investigated and superior experimental results are shown on several datasets as compared to the baseline.

**Strengths:**

The introduction of the local curvatures for structural encoding in graph neural networks is the key contribution of the paper.  A theoretical result (Theorem 1) is also established suggesting improved expressivity due to LCP. However the result is rather qualitative without a quantitative characterization of the extent to which the expressivity is improved. Thus the theoretical development is rather light.

Overall the paper is very well written and, for most parts, easy to read.

The idea is sound and the experiments look convincing to this reviewer.

**Weaknesses:**

I do not see an obvious weakness in the paper, just like I do not see its development particularly striking.  To me, the paper falls into those works that have a sound intuitive idea, which is validated via empirical evaluation. The paper does not appear to touch on the studied problem (i.e., the issues of over-smoothing and over-squashing) at a fundamental level or at depth. But it is perhaps above the acceptance threshold.

**Questions:**

N/A

---

> ### Author Response · Authors · 2023-11-15
>
> We thank the reviewer for the encouraging feedback.
>
> > A theoretical result (Theorem 1) is also established suggesting improved expressivity due to LCP. However the result is rather qualitative without a quantitative characterization of the extent to which the expressivity is improved. Thus the theoretical development is rather light.
>
> We would like to note that the WL hierarchy is by now a standard tool for measuring expressivity, which is widely used in the Graph Machine Learning literature. Theorem 1 allows for categorizing LCP in terms of the WL hierarchy.
>
> > The paper does not appear to touch on the studied problem (i.e., the issues of over-smoothing and over-squashing) at a fundamental level or at depth.
>
> We have added additional plots depicting the (normalized) Dirichlet energy, which is commonly used to measure over-smoothing, in Appendix A.9. We find that using the LCP increases the Dirichlet energy almost as much as ORC-based rewiring (BORF), which was explicitly designed to deal with over-smoothing.

---

> > ### Comment · Reviewer_GHx9 · 2023-11-21
> > **I have read your comments**
> >
> > Thank you for the additional information.

---

### Official Review · Reviewer_JWiJ · 2023-11-01

**Soundness:** 3 good
**Presentation:** 3 good
**Contribution:** 3 good
**Rating:** 8
**Confidence:** 2

**Summary:**

This paper proposes improving graph neural networks, e.g. graph convnets. The idea is to encode structural information through Local Curvature Profiles, enabling each node to better characterize the geometry of its neighborhood. Instead of rewriting the graph, the proposed approach adds summary statistics about each node's local curvature to the features of each node.

On a variety of different tasks, this approach improves the performance of the resulting graph neural nets.

**Strengths:**

This paper seems reasonable to this reviewer, outperforming baseline encoding approaches or approaches that require rewiring.

Perhaps most surprising to this reviewer is that it improves performance of GATs (seemingly similar to transforms in that they use self-attention?) as it would seem reasonable that such a network would be able to dynamically compute something similar to these statistics.

The experiments seem reasonably done at least to this reviewer (not an expert in this area at all), involving both LCP itself as well as combining it with positional encoding, and then later rewiring.

**Weaknesses:**

It's not obvious to this reviewer what the weaknesses are. The main concern to this reviewer is that some large pretrained transformer could do better than any of the proposed methods, but that's a very general concern these days. Possibly this approach or GNNs in general could work better on more specialized tasks where there are a very large number of nodes.

**Questions:**

How do the results compare versus model size? E.g. could making a GCN deeper allow it to implicitly compute these kinds of features itself? What's stopping it from doing that?

---

> ### Author Response · Authors · 2023-11-15
>
> We would like to thank the reviewer for the careful read of our submission and for the very positive feedback.
>
> > How do the results compare versus model size? E.g. could making a GCN deeper allow it to implicitly compute these kinds of features itself? What's stopping it from doing that?
>
> As shown in [1], message-passing GNNs are generally unable to detect whether a graph contains a cycle of a specific length. The number of cycles of lengths 3-5 is, however, crucial when computing Augmentations of Forman’s Ricci Curvature (see Appendix A.1), which according to [2] can be thought of as low-level approximations of the Ollivier-Ricci Curvature. As such, we would not expect a message-passing GNN to be able to implicitly compute curvature-based features, even if we increased the model size. Even if this were possible, increasing the model size - especially the depth - to such a degree would most likely result in over-smoothing, thus degrading performance.
>
> > Perhaps most surprising to this reviewer is that it improves performance of GATs (seemingly similar to transforms in that they use self-attention?) as it would seem reasonable that such a network would be able to dynamically compute something similar to these statistics.
>
> In general, our reasoning in the previous comment still applies as GAT is a message-passing GNN. However, we agree with the reviewer that it is surprising that GAT does not at least compute some form of approximation. This phenomenon has also been observed in recent works on over-squashing [3] and presents a promising avenue for future research.
>
> [1] Loukas, Andreas. "What graph neural networks cannot learn: depth vs width." arXiv preprint arXiv:1907.03199 (2019).
>
> [2] Jost, Jürgen, and Florentin Münch. "Characterizations of Forman curvature." arXiv preprint arXiv:2110.04554 (2021).
>
> [3] Di Giovanni, Francesco, Lorenzo Giusti, Federico Barbero, Giulia Luise, Pietro Lio, and Michael M. Bronstein. "On over-squashing in message passing neural networks: The impact of width, depth, and topology." In International Conference on Machine Learning, pp. 7865-7885. PMLR, 2023.

---

> > ### Comment · Reviewer_hevh · 2023-11-23
> >
> > Dear Authors,
> >
> > Thank you for the response to my comments. The rebuttal has addressed my concerns and I'll keep my score.

---

> > ### Comment · Reviewer_JWiJ · 2023-11-23
> > **thanks! will keep my score**
> >
> > thanks for the response! I'm still on board with accepting this paper so I'd like to keep my score.

---

### Official Review · Reviewer_hevh · 2023-11-01

**Soundness:** 3 good
**Presentation:** 3 good
**Contribution:** 3 good
**Rating:** 6
**Confidence:** 3

**Summary:**

The paper addresses the crucial issue of improving the performance of GNNs through structural encodings. The authors present a novel approach based on discrete Ricci curvature, termed Local Curvature Profiles (LCP), and demonstrate its significant effectiveness in enhancing GNN performance. They also investigate the combination of local structural encodings with global positional encodings and compare these encoding types with curvature-based rewiring techniques. The paper makes important contributions to the field of Graph Machine Learning and provides valuable insights into the potential of curvature-based encodings.

==================================

Update: I appreciate the authors for answering my questions and providing more experimental results. I would like to keep my scores.

**Strengths:**

- LCP provides a unique way to encode the geometry of a node's neighborhood, and the paper convincingly demonstrates its superior performance in node and graph classification tasks.

- The paper investigates the combination of local structural encodings with global positional encodings, showing that they capture complementary information about the graph. This finding is valuable as it suggests that using a combination of different encoding types can result in enhanced downstream performance. The authors provide empirical evidence to support this claim.

- A theoretical analysis of LCP's computational efficiency and its impact on expressivity is included in the paper.

**Weaknesses:**

- Some parts of the introduction are a bit dense and may be challenging for readers not deeply familiar with the field. A clearer presentation of the background and motivation could benefit a wider audience.

- Including experiments on a more diverse set of datasets and domains would be better.

**Questions:**

How well does LCP generalize across different domains, and what factors might influence its applicability in practical scenarios?

Are there any computational bottlenecks when implementing LCP in large-scale graph datasets, and what strategies or optimizations could be considered to address these issues?

---

> ### Author Response · Authors · 2023-11-15
>
> We would like to thank the reviewer for the encouraging feedback.
>
> > Including experiments on a more diverse set of datasets and domains would be better.
>
> We have added four additional datasets in appendix sections A.7 and A.8: three heterophilious node classification datasets [1] and one graph regression task [2]. These datasets contain both social networks (e.g. Amazon ratings) and molecules (Zinc), i.e. vary in the domain and graph topology. We find that the LCP outperforms all other encodings considered and increases performance for all three models (GCN, GIN, GAT).
>
> > How well does LCP generalize across different domains, and what factors might influence its applicability in practical scenarios?
>
> Our experiments include social networks and networks relevant to the natural sciences (e.g. molecules). While we find that the LCP leads to considerable performance gains across domains, it seems that it is generally more useful for graph-level tasks (classification, regression) than for node-level tasks.
>
> > Are there any computational bottlenecks when implementing LCP in large-scale graph datasets, and what strategies or optimizations could be considered to address these issues?
>
> The main computational bottleneck that comes with using the LCP is that it scales cubically with the maximum degree of a node in the graph. For example, for the Tolokers dataset that we have added, computing the LCP is infeasible because of the high average degree in the graph (it takes more than six hours, which we consider a timeout).
>
> We can address this issue by using the combinatorial approximations presented in appendix A.1.3 instead of the ORC itself. These approximations scale linearly in the max degree, so computing the (approximate) LCP for Tolokers, for example, now only takes a few seconds. In addition, the LCP can be implemented with a different curvature notion (Forman’s curvature, short: FRC), which is more scalable than ORC. Experiments for this can be found in Table 5.
>
> > Some parts of the introduction are a bit dense and may be challenging for readers not deeply familiar with the field. A clearer presentation of the background and motivation could benefit a wider audience.
>
> We will improve the clarity of the writing in the introduction and background and related works sections. If the reviewer has concrete suggestions on which concepts a wider audience might not be familiar with, we would be grateful.
>
> [1] Platonov, Oleg, Denis Kuznedelev, Michael Diskin, Artem Babenko, and Liudmila Prokhorenkova. "A critical look at the evaluation of GNNs under heterophily: are we really making progress?." arXiv preprint arXiv:2302.11640 (2023).
>
> [2] Gómez-Bombarelli, Rafael, Jennifer N. Wei, David Duvenaud, José Miguel Hernández-Lobato, Benjamín Sánchez-Lengeling, Dennis Sheberla, Jorge Aguilera-Iparraguirre, Timothy D. Hirzel, Ryan P. Adams, and Alán Aspuru-Guzik. "Automatic chemical design using a data-driven continuous representation of molecules." ACS central science 4, no. 2 (2018): 268-276.

---

### Author Response · Authors · 2023-11-15
**General response**

We thank all reviewers for their careful read of our submission and the detailed feedback.

In the revised manuscript, we have addressed all reviewers’ comments, changes are highlighted in red font. We believe that the revisions following the reviewers’ feedback have significantly improved the paper.

We briefly summarize the key points of our rebuttal, before responding to each reviewer’s comments in detail:

- We have significantly expanded the experimental section by including additional baselines on a wider range of graph domains and topologies. In particular, we added several baselines from the heterophilious graphs dataset [1] for node classification, as well as the ZINC dataset [2] for graph regression. The additional experiments confirm the performance gains observed in our original experiments.
- We have expanded the ablation experiments for the choice of summary statistics included in LCP. The additional experiments confirm that our original configuration yields the highest accuracy in almost all cases.
- To clarify how LCP addresses over-smoothing, we have included additional ablations using the (normalized) Dirichlet energy.
- Following reviewer feedback, we have added additional explanation of notation and implementation details to improve the clarity of the paper.

[1] Platonov, Oleg, Denis Kuznedelev, Michael Diskin, Artem Babenko, and Liudmila Prokhorenkova. "A critical look at the evaluation of GNNs under heterophily: are we really making progress?." arXiv preprint arXiv:2302.11640 (2023).

[2] Gómez-Bombarelli, Rafael, Jennifer N. Wei, David Duvenaud, José Miguel Hernández-Lobato, Benjamín Sánchez-Lengeling, Dennis Sheberla, Jorge Aguilera-Iparraguirre, Timothy D. Hirzel, Ryan P. Adams, and Alán Aspuru-Guzik. "Automatic chemical design using a data-driven continuous representation of molecules." ACS central science 4, no. 2 (2018): 268-276.

---

### Meta-Review · Area_Chair_bMuc · 2023-12-15

**Metareview:**

The paper introduces Local Curvature Profiles (LCP) as a method for encoding the geometry of a node's neighborhood, showcasing superior performance in node and graph classification tasks. The authors show that combining local structural encodings with global positional encodings is complementary, suggesting that employing different encoding types can enhance downstream performance, a claim supported by empirical evidence.

The inclusion of a theoretical analysis delves into LCP's computational efficiency and its impact on expressivity, presenting a convincing argument for its superiority over baseline approaches or methods requiring rewiring. Notably, the paper's method improves the performance of Graph Attention Networks (GATs), which share similarities with transformers using self-attention.

The experiments, involving LCP, positional encoding, and later rewiring, are deemed reasonable. The key contribution of introducing local curvatures for structural encoding in graph neural networks is acknowledged, along with a qualitative theoretical result (Theorem 1) suggesting improved expressivity due to LCP. However, some reviewers noted that the theoretical results are not very significant and insightful.

Overall, the paper is well-written, presents a sound and novel method, and convincing experimental results.

Only one of the reviewers was very negative about the paper. However, considering that the criticism was shallow and that the reviewer did not engage at all iduring the rebuttal, the AC decided to ignore said review.

**Justification For Why Not Higher Score:**

Some weaknesses were pointed out (weak, shallow theoretical results).

**Justification For Why Not Lower Score:**

Three of the four reviewers are positive about the paper.

---

### Decision · Program_Chairs · 2024-01-16

Accept (poster)